# Significant Decrease in Childhood Obesity and Waist Circumference over 15 Years in Switzerland: A Repeated Cross-Sectional Study

**DOI:** 10.3390/nu11081922

**Published:** 2019-08-15

**Authors:** Isabelle Herter-Aeberli, Ester Osuna, Zuzana Sarnovská, Michael B. Zimmermann

**Affiliations:** ETH Zurich, Institute of Food, Nutrition and Health, Laboratory of Human Nutrition, 8092 Zurich, Switzerland

**Keywords:** childhood obesity, waist circumference, Switzerland, overweight, skinfold thickness, physical activity

## Abstract

Despite a global increase in childhood obesity over the past decades, several countries, including Switzerland, have recently reported stabilizing trends. Using national data from school-aged children in Switzerland over the past 16 years, our study aim was to assess changes in the prevalence of overweight and obesity, central adiposity and predictors of obesity related to lifestyle and parental factors. Nationally representative samples of children aged 6–12 years were studied in 2002 (*n* = 2493), 2007 (*n* = 2218), 2012 (*n* = 2963), and 2017/18 (*n* = 2279). Height and weight, waist circumference, and multiple skinfold thicknesses were measured. Potential risk factors for overweight and obesity were determined using a self-administered questionnaire in 2017/18, collecting data on diet, physical activity, and parental factors. Prevalence (95% CI) of overweight (incl. obesity) and obesity in 2017/18 was 15.9% (14.4–17.4) and 5.3% (4.5–6.3), respectively. Binary logistic regression revealed a small but significant decrease in the prevalence of overweight (including obesity) since 2002 (OR (95% CI) = 0.988 (0.978–0.997)), while the change in obesity alone was not significant. The most important risk factors for childhood overweight/obesity in 2017/18 were low parental education, non-Swiss origin of the parents, low physical activity of the child, and male sex. In conclusion, we have shown a small but significant declining trend in the childhood overweight/obesity prevalence over the past 15 years in Switzerland. Based on the risk factor analysis, preventive action in schoolchildren might be most effective in boys, migrant populations, and families with lower education, and should emphasize physical activity.

## 1. Introduction

Globally, non-communicable diseases (NCDs) are an important public health concern responsible for substantial mortality and morbidity and causing high socio-economic costs [1]. There is evidence that the major risk factors for NCDs can be associated with behavioral patterns mainly established during childhood and adolescence, which then continue into adulthood [2,3,4]. Moreover, the onset of many NCDs, such as obesity, diabetes, and cardiovascular diseases (CVD), can be reduced/prevented when addressing risk factors early in life [5]. One example of such a risk factor is childhood obesity, which is strongly associated with adult NCDs [2,4].

Worldwide, the prevalence of overweight and obesity in children has increased dramatically during the last decades [6,7]. In the US, the prevalence of obesity increased from 8% to 14% in 6–12 year old children between 1976 and 1994 [8] and even reached 18.4% in 2015/16 [9]. The prevalence of childhood obesity continues to increase globally, especially in certain countries and subgroups [10]. Countries with an emerging economy, such as China, where childhood obesity increased from 5.3% in 1995 to 20.5% in 2014, are especially affected [11]. Vulnerable subgroups vary between countries. While in high-income countries, such as the UK, children from a low socioeconomic background have a higher obesity risk [12], an inverse trend is currently still observed in low-income countries [11]. However, recent projections from the Organization for Economic Co-operation and Development (OECD) show an overall steady increase of obesity rates, even in OECD countries (which do not include most developing and transition countries), until at least the year 2030 [13]. Nevertheless, since 2007, data from several countries (e.g., from Australia, China, France, and the USA) suggests that the increase in obesity prevalence has slowed substantially, or even leveled off [14]. In our last Swiss nationally representative study in 2012, we reported a stabilized, combined prevalence of overweight and obesity between 1999 and 2012 at approximately 19% [15]. Whether this stabilization will prove to be a turning point or just a transient trend is still uncertain. Most importantly, even if some countries may have achieved a stabilizing or even decreasing trend over the past decade or two, global trends between 2010 and 2016 still indicate an increasing overweight and obesity prevalence in all WHO regions, ranging from +9% in the Americas to +48% in the South-East Asian region [16]. With a total of 340 million children between 5 and 19 years of age who are at risk for poor health both during childhood and in adulthood, this represents an immense burden for health care services [16].

The causes of obesity are multifactorial. Changes in food habits and an increasingly sedentary lifestyle, as well as genetic, social, and cultural factors, are important determinants of population patterns of obesity. Lifestyle interventions addressing different treatment aspects are generally recommended for the treatment of childhood obesity [17,18,19]. Although randomized controlled trials concerning weight loss have shown some success, weight loss is often difficult to sustain in the long term [17]. Therefore, the prevention of obesity should be emphasized rather than its treatment. In order to develop more targeted and successful prevention strategies, a better understanding of the predictors of childhood obesity is important. There are different stages at which the development of childhood weight status can be influenced. Early life factors, such as maternal nutrition and weight gain during pregnancy, breastfeeding, and complementary feeding have been shown to be important predictors [20]. However, other risk factors, more related to the current lifestyle, such as current eating behavior and physical activity, as well as parental socioeconomic status, have also been shown to be important determinants [20,21,22,23].

The principle objective of this study was to investigate the trend of the prevalence of overweight, obesity, and central adiposity in 6 to 12 year old schoolchildren in Switzerland between 2002 and 2017/18. The secondary objective was to assess predictors for the development of obesity during childhood using a self-administered questionnaire. Furthermore, we compared the sensitivity of different reference curves for the determination of Swiss childhood obesity.

## 2. Subjects and Methods

### 2.1. Study Design

Data for this analysis was obtained from four nationally representative, cross-sectional, school-based surveys in Switzerland. Detailed descriptions of the studies conducted in 2002, 2007, and 2012 have been published previously [15,24,25,26]. In all four surveys, we used identical stratified probability-proportionate-to-size (PPS) cluster sampling to obtain a nationally representative sample of 6–12 year old children in Switzerland. Based on current census data, the Swiss Federal Office of Statistics divided Switzerland into five geographic regions: west (French language), north-west (German language), north-east (German language), central east (German language), and south (Italian language). Each of these regions was divided into three strata by population size of the communities (small communities: <10,000 inhabitants; middle-sized communities: 10,000–100,000 inhabitants; large communities >100,000 inhabitants). Based on the list of communities, we applied two-stage PPS random cluster sampling to obtain independent national samples of the required population group. We thus randomly selected 60 communities and per community, one school, across Switzerland. The aim in all surveys was to sample at least one in 250 children in Switzerland.

### 2.2. Enrollment and Participation

Participation was voluntary in 2002, 2007, and 2017/18; written informed consent was obtained from all parents or legal guardians of the participating children [24,26,27]. The enrollment process was different in 2012, using passive consent where parents had the possibility to withdraw their child from participation [15]. For all surveys, an information letter describing the study and the examination process was sent to the school principal, teachers, parents, and children two to four weeks prior to the measurement day in order to give parents/children sufficient time to consider participation. For the 2017/18 survey, data were collected from September 2017 to March 2018. All children attending the selected classes and aged between 6 and 13 years were allowed to participate in the study.

In 2017/18, ethical approval was obtained from the Cantonal Ethical Committee Zurich as a representative of all other Cantonal Ethical Committees (BASEC-Nr. 2017-01202). Where needed, we obtained local authorization from cantonal or communal school or health departments. The cantons of Vaud and Fribourg decided not to participate in the survey. The study was registered on clinicaltrials.gov (NCT03309189). The previous studies were approved by the relevant authorities, as previously described [15,24,26,27].

### 2.3. Anthropometric Measures

Measurement procedures for weight and height, waist circumference (WC), and skinfold thicknesses (SFT) were identical for all four surveys, but WC was not measured in 2002 and SFTs were not available in 2007, as the caliper used was found to be unreliable half way through the study. A detailed description of the procedures for 2017/18 is given in the following paragraph. Except for the types of scale and stadiometer, this description is also valid for the previous surveys. Children always left the classroom in pairs and examinations were carried out in a separate room by two trained examiners. For all subjects, body weight was measured to the nearest 0.1 kg using a digital balance (Beurer GmbH, GS203 wood) and height was measured to the nearest 0.1 cm using a transportable stadiometer (SECA, 213). Body Mass Index (BMI) was calculated as weight divided by height squared. WC was further measured in all children midway between the lowest rib and the iliac crest using a non-stretchable measuring tape in duplicate.

The body fat percentage (BF%) of each child was determined by measuring SFTs at four sites using a Harpenden Skinfold Caliper with a resolution of 0.2 mm. The four sites were triceps, biceps, subscapular, and suprailiacal [28]. For the triceps, the mid-point of the back of the upper arm between the tip of the olecranon and the acromial process was determined by measuring with the arm flexed at 90 degrees. With the arm hanging freely at the side, the caliper was applied vertically above the olecranon at the marked level. Over the biceps, the SFT was measured at the same level as the triceps, with the arm hanging freely and the palm facing straight in the frontal plane. At the subscapular site, the skinfold was picked up just below the inferior angle of the scapula at 45° to the vertical along the natural cleavage lines of the skin. The suprailiac SFT was measured above the iliac crest, just posterior to the midaxillary line and parallel to the cleavage lines of the skin, with the arm lightly held forward. All sites were measured on the right side of the body in duplicate. In every tenth participant, measurements of SFT and WC were conducted by two different investigators to determine inter-observer variability, while in all other participants, duplicate measurements were performed by the same investigator to determine intra-observer variability. The mean of both values was used for later data interpretation. Technical error of measurement (TEM) was used to evaluate inter- and intra-observer variability and was calculated as follows [29]:
$$\mathrm{TEM}=\sqrt{(\Sigma D^{2}}/2N)$$
where *D* is the difference between the two measurements and *N* is the number of individuals measured. For better comparability between variables, relative TEM (%TEM) was then calculated as follows [29]:%TEM = (TEM/mean) × 100

### 2.4. Assessment of Predictors for Obesity

A questionnaire (hard copy) was distributed to each participating child inquiring about socioeconomic background (parental education and birth place of parents), general health, physical activity, and nutritional habits. The questionnaire was available in three national languages: German, French, and Italian. The questionnaire contained a question on the place of birth of both parents of the child. For the analysis, the place of birth of the parents was categorized as follows: ‘Both Swiss’, Swiss and non-Swiss’, and ‘Both non-Swiss’. The education level of both parents was further assessed using the following categories: ‘obligatory school time (=11 years, including kindergarten)’, ‘apprenticeship without general qualification for university entrance’, apprenticeship with general qualification for university entrance’, ‘university of applied sciences or technical university’, and ‘university’. The educational levels of both parents were combined and categorized into the following three groups: ‘low’ (obligatory school time), ‘moderate’ (apprenticeship with or without professional maturity), and ‘high’ (university of applied sciences, technical university, or university).

Regarding physical activity, the children were asked how many days in a typical week they are physically active for at least 60 min (not differentiating between physical activity levels). Answers were categorized as follows: ‘≤1 day/week’, ‘2–3 days/week’, ‘4–5 days/week’, and ‘≥6 days/week’. The children were asked how much time they spend watching TV or videos; playing on the computer; using a cell phone, tablet, or similar device; using social media; and completing homework on the computer. The times for all activities were summed up to cover the overall screen time. Screen time was then categorized into ‘≤1 h/day’, ‘>1 h and ≤2 h/day’, ‘>2 h and ≤3 h/day’, and ‘>3 h/day’.

Several dietary factors were assessed using the questionnaire. The children were asked how many times in the last four weeks they consumed the following items: soft drinks, fruit and vegetable juices, fruits, vegetables (including salad), milk and dairy products, meat, and fish. The answers given were categorized as follows. For soft drinks, as well as meat and fish: ‘≤1 day/week’, ‘2–4 days/week’, ‘5–6 days/week’, and daily. For fruit and vegetable juices, fruits, and vegetables: ‘<1 time/day’, ‘1–2 times/day’, ‘3–4 times/day’, and ‘≥5 times/day’. For milk and dairy products: ‘≤1 day/week’, ‘2–4 days/week’, ‘5–6 days/week’, ‘1 time/day’, and ‘>1 time/day. Children were further asked about their breakfast habits and could choose from the following options: eating breakfast ‘daily’, ‘sometimes (e.g., only on weekends)’, and ‘never’.

To assess the general health status of the children, a few further questions were asked. Based on the time they usually go to sleep and get up, the mean sleep duration was calculated. Furthermore, the children were asked whether they suffered from any disease (diabetes, asthma, other chronic disease) and how they felt about their health in general (feeling very well, well, rather well, or not well). The last two questions asked them to judge their weight status (weight perception: much too thin, too thin, about right, a little too heavy, or much too heavy) and their life in general on a scale from 1 to 10 (life satisfaction). These questions were based on the questionnaire used by the international study Health Behaviour in School-Aged Children (HBSC).

### 2.5. Data Analysis

Data and statistical analysis was conducted using IBM SPSS Statistics 24 (IBM Company, Armonk, NY, USA) and Excel (Microsoft Office 2016, Microsoft Corporation, Redmond, WA, USA). Using the BMI data, the prevalence of overweight and obesity in 6–12 year old children in Switzerland was calculated based on the Centers for Disease Control and Prevention (CDC) reference values using the cut offs of the 85th and the 95th percentiles [30], the International Obesity Task Force (IOTF) reference values for overweight and obesity extrapolated from adult cut-off points [31], and the 85th and the 95th percentiles of the WHO BMI for age curves [32]. Underweight children were always included in the healthy weight category. Using Swiss reference values for WC, the 90th percentile was defined as a cut-off for central adiposity [33]. Using the mean value of the repeated SFT measurements, the body density and BF% were calculated using the following equations [34]:

BF% = (562 − 4.2 × [Age (y) − 2])/*D* − (525 − 4.7 × [age(y) − 2])

where *D* = body density.

For boys: *D* (g/mL) = 1.169 − 0.0788 × log10 (sum of 4 SFT [mm])

For girls: *D* (g/mL) = 1.2063 − 0.0999 × log10 (sum of 4 SFT [mm])

Swiss reference values for BF% were used to determine overweight and obesity based on this measurement. The 85th percentile was used for overweight and the 95th for obesity [27].

The sensitivity and specificity of the three BMI reference curves in comparison to BF% references were calculated. Based on the results of our previous study in 2002 [27], we expected the CDC reference curves to perform better than those obtained by IOTF. However, in the meantime, another set of reference curves from WHO became available, so we decided to redo the sensitivity and specificity analysis using all three sets of references. The CDC references showed a sensitivity of 74.4% and 77% and a specificity of 93.9% and 97.1% for overweight, including obesity, and obesity, respectively. The corresponding values for the IOTF references were a sensitivity of 74.1% and 60.8% and a specificity of 94.9% and 98.6% for overweight, including obesity, and obesity, respectively. The WHO references showed a sensitivity of 84.3% and 79.7% and a specificity of 88.4% and 96.4% for overweight, including obesity, and obesity, respectively. Based on the reasonably high sensitivity (ca. 25% false negative) and the high specificity (ca. 5% false positive), we decided to use the CDC reference values for data presentation and statistical analysis in this study.

The prevalence of overweight and obesity between sex, age groups, geographic and demographic regions, and parental characteristics was compared using the chi-square test followed by a z-test to check for significant differences between individual values (Bonferroni correction). Similarly, the prevalence of overweight and obesity between regions and communities of different sizes was conducted using the chi-square test followed by a z-test (including Bonferroni correction for multiple comparisons). The 95% confidence intervals for all prevalences were calculated using the Wilson procedure [35], as described by Robert Newcombe [36]. A binary logistic regression was used on the trends in overweight and obesity prevalence between 2002 and 2017/18, with survey year as a continuous variable.

Data entry for the questionnaire was done using a standardized procedure by five trained persons (EO, SG, LH, SB, and ZS). Multinomial logistic regressions were used to examine the associations between BMI category (by using the CDC references) and potential predictors. In a first step, each individual variable was tested (dependent variable: BMI category; factor: each of the potential predictors individually). In a second step, all factors showing a significant association in the univariate model were added to the multivariate, stepwise, backward model, controlling for age (dependent variable: BMI category; factors: all significant predictors; covariate: age). All models were checked for model fitting and parameters were only included in the final model if the likelihood ratio test showed *p* < 0.05. *p* values <0.05 were considered significant for all analyses.

Children were included in the analysis if at least age, weight, height, and waist circumference were available. If SFT could not be measured, the remaining data of those children was still included in the analysis. For the determination of predictors of obesity, only those children who returned the questionnaire were included.

## 3. Results

### 3.1. Recruitment Statistics of the Obesity Survey 2017/18

We contacted a total of 491 schools in order to recruit the required 60 schools, which resulted in a response rate of 12.2% at the school level. In the consenting schools, we invited 4165 children to participate in the study, of which 2382 consented. On the day of measurement, 90 children were absent, resulting in a sample size of 2292 and a response rate within participating schools of 55%. Of this number, we excluded data of another 13 children because their age was either below 6 years or 13 years and above. Therefore, the total number of participants included for data analysis was 2279. This corresponds to one in 212 children in this age group in Switzerland. For eight children, SFT measurements were not available due to an injury at one of the measurement sites, usually the arm. The remaining data for those children were included in the analysis. There was a slight underrepresentation of the Western region (12 out of 15 schools) because two cantons decided not to participate in the study. The missing schools were replaced by two schools in the Northeastern region (20 instead of 18 schools) and one in the Central and Eastern region (15 instead of 14 schools).

### 3.2. Inter- and Intra-Observer Variability in the Obesity Survey 2017/18

The inter-observer variability, calculated as %TEM, was 1.5% for WC and ranged between 7.5% and 11.4% for the four SFT measurements, with the highest value for the suprailiacal measurement. The intra-observer variability, also calculated as %TEM, was 1.3% for WC and ranged between 2.5% and 4.5% for the four SFT measurements, with the highest values also for the suprailiacal measurement.

### 3.3. Trends in Overweight and Obesity Prevalence between 2002 and 2017/18

The participant characteristics of all four studies are shown in Table 1. Prevalence and trends in the development of overweight and obesity, defined using the CDC references, between 2002 and 2017/18, are shown in Table 2 and Figure 1. For comparison, using the WHO references for the 2017/18 data, the prevalence of overweight alone was 17.3% (15.2–19.6) in boys and 14.8% (12.9–17.0) in girls, while the prevalence of obesity was 7.3% (6.0–9.0) in boys and 4.7% (3.6–6.1) in girls. The development of central adiposity based on WC between 2007 and 2012 is shown in Table 2.

Using a binary logistic regression over the years 2002 to 2017/18, a weak but significant trend towards a reduction in childhood overweight including obesity could be identified (OR (95% CI) = 0.988 (0.978–0.997)). Similarly, the prevalence of overweight alone showed a significant decreasing trend (OR = 0.986 (0.975–0.997)), whereas there was no change in the prevalence of obesity (OR = 0.994 (0.979–1.010)). When grouped by sex, a significant decreasing trend in overweight including obesity could be observed in girls (OR = 0.986 (0.973–1.000)), but not in boys (OR = 0.989 (0.976–1.002)). The trend was not significant for overweight alone (OR = 0.985 (0.970–1.001) for girls and OR = 0.986 (0.971–1.002) for boys) or obesity alone (OR = 0.991 (0.968–1.014) for girls and OR = 0.996 (0.976–1.017) for boys). To investigate whether the changes affected predominantly younger or older children, the entire group was divided into 6–9 year old and 10–12 year old children. When comparing the younger (6–9 years) to the older (10–12 years) children, a significant trend towards a reduction in overweight including obesity could be seen in the younger, but not the older, children (OR = 0.984 (0.972–0.996) for 6–9 year old children and OR = 0.991 (0.977–1.005) for 10–12 year old children). Similarly, for overweight alone, a significant decrease could be seen for the younger children (OR = 0.983 (0.968–0.998)), but not the older ones (OR = 0.988 (0.971–1.005)). For the prevalence of obesity alone, no significant change could be shown for either age group (OR = 0.990 (0.970–1.010), for 6–9 year olds and OR = 0.998 (0.975–1.022) for 10–12 year olds). Similar to the prevalence of overweight and obesity, central adiposity based on WC showed a weak but significant decrease between 2007 and 2017/18 (OR = 0.949 (0.932–0.966)).

The prevalence of overweight including obesity was significantly higher in boys compared to girls only in the 2007 survey (*p* < 0.05), while the prevalence of obesity was significantly higher in boys compared to girls in the years 2007, 2012, and 2017/18 (*p* < 0.05). The prevalence of overweight alone, on the other hand, did not differ between sex in any of the surveys. Significant differences in the three prevalence estimates between years are shown in Table 2. The prevalence of the increased risk for metabolic disease based on WC did not differ between boys and girls in any of the survey years (Table 2).

### 3.4. Predictors for the Development of Overweight and Obesity in 2017/18

The prevalence of overweight ranged from 8.4% (6.4–11.0) in the Central and Eastern region to 12.3% (9.7–15.4) in the Western region, while the prevalence of obesity ranged from 4.6% (2.8–7.6) in the Northcentral region to 7.8% (4.1–14.1) in the Southern region. However, none of the regional differences reached significance (*p* > 0.05). The prevalence of overweight was significantly higher in large cities (>100,000 inhabitants, 15.9% (11.8–21.1)) compared to smaller communities (<10,000 inhabitants, 9.7% (8.2–11.3), *p* < 0.05), with medium-sized communities in between (10.5% (8.5–13.1)). On the other hand, there was no difference in obesity prevalence related to community size (small: 4.6% (3.6–5.9), medium: 6.8% (5.2–9.0), large: 5.0% (2.9–8.6), *p* > 0.05). When grouping children by age (6–9 years and 10–12 years), neither overweight nor obesity prevalence differed significantly between the groups (*p* > 0.05, data not shown).

Potential predictors for overweight and obesity were assessed using a questionnaire as described above. The questionnaire was returned by 2149 children (94.3%). An overview of the answers (frequency (%)) by weight status group is given in Appendix A.

The prevalence of overweight was significantly higher in children whose parents were both born outside of Switzerland compared to those with two parents born in Switzerland (*p* < 0.05). The prevalence of obesity, however, was higher for both children with one or two parents born outside of Switzerland compared to those with two parents born in Switzerland (*p* < 0.05) (Figure 2A). The countries most often named as the birthplace of mothers and fathers combined were Portugal, Germany, Kosovo, Italy, and Macedonia. Parental education was a significant predictor of the weight status of children. The prevalence of overweight was significantly lower in children of parents with a high education level compared to those with a medium education level (*p* < 0.05). The prevalence of obesity was highest in children of parents with a low education level, lowest in those of parents with a high education level, and differed significantly between all three groups (*p* < 0.05) (Figure 2B). There were significantly more parents with low education who were both born outside of Switzerland then who were both Swiss, while there were significantly less parents with high education who were both born outside of Switzerland than who were both Swiss. Sleep duration was significantly higher in healthy weight children (10.49 h/day) compared to overweight (10.37 h/day, *p* = 0.028 vs. HW) and obese (10.25 h/day, *p* = 0.001 vs. HW) children, but did not differ between overweight and obese children (*p* = 0.312).

Logistic regression models were used in two steps to investigate the effect of the different potential predictors on weight status in 2017/18, as described earlier. The factors showing a significant association with weight status (as defined using the CDC BMI references) in the individual models, and thus included in the multiple model, were parental origin, parental education, screen time, physical activity, sleep duration (tertiles), eating breakfast, and vegetable consumption. The individual model could not be calculated for several factors due to very small numbers in some of the categories: Soft drinks, fruit/vegetable juice, milk and dairy products, meat and fish, any disease, general health, life satisfaction, and self-perception of weight. Furthermore, the individual model was not significant for fruit consumption. The results of the stepwise multiple regression, which, besides the factors mentioned above, also included sex and age as covariates, are shown in Table 3. Only parental education, parental origin, physical activity, and sex remained in the final model as significant predictors of overweight and/or obesity. Based on the pseudo R-Square (Naglekerke’s adjusted value), however, the model only explains 11.7% of the variability in overweight and/or obesity.

## 4. Discussion

For the first time since nationally representative surveys on childhood obesity have been conducted in Switzerland, we have found a small but significant decrease in prevalence in a time-trend analysis between 2002 and 2017/18. This trend was not only seen for overweight/obesity defined using BMI, but also when using WC to identify children with an increased risk for metabolic disease. Nevertheless, the decrease, even though statistically significant, was only small, with an OR of 0.988. Despite this, the findings confirm our earlier results from 2012, where we demonstrated a stabilization of the prevalence [15]. A similar stabilizing trend with some fluctuations was shown in several countries over the past two decades [13,14,37]. A large German study recently reported a stabilizing, and in some age groups, even decreasing, trend in overweight and obesity prevalence between 2005 and 2015 [38]. In a report based on the WHO European Childhood Obesity Surveillance Initiative (COSI) surveys 2007/2008 and 2009/2010, some countries showed an increase in prevalence, while others showed a decrease over the same period of time [39]. However, this is not necessarily in conflict with our results and those of several other countries. As the time frame for the WHO comparison was only two years, short-term fluctuations in both directions were registered rather than a longer term trend. In Switzerland, a similar trend as in the current study was already demonstrated by the BMI monitoring program of school physicians of the three large cities Bern, Basel, and Zurich; in the age group comparable to our current survey (6–12 years), they reported an overall prevalence of overweight including obesity of 22% between 2005 and 2009 and 21% between 2013 and 2016, using IOTF reference values [40].

Several factors have likely contributed to the decrease or stabilization in childhood obesity in Switzerland. Several hypotheses have been discussed that might be responsible for such a trend [14]. One possibility is that childhood obesity has been recognized as a major public health concern and thus healthy eating and increased physical activity have been promoted in national or school-based campaigns and interventions. In Switzerland, large campaigns have been conducted over the past decades, with the most important one being the national program on nutrition and physical activity, which was run between 2008 and 2016 [41]. A second possibility, the saturation equilibrium hypothesis, explains a stabilization of obesity rates by reaching a point of saturation at which any child predisposed to becoming overweight has become overweight and the remaining children are unaffected by a given obesogenic environment [14]. A third possibility is that with increasing public awareness, overweight children or their parents are less likely to participate in studies on this topic. We had a lower participation rate (55%) in the current study compared to 73% to 95% in earlier studies, so we cannot rule out a potential selection bias. However, to try and minimize this, we did not mention the words ‘overweight’ or ‘obesity’ in the participation information. The study was presented as a health and nutrition survey, where anthropometric measures were taken along with the assessment of data on diet and physical activity. Moreover, a previous study reported that obtaining active or passive parental consent resulted in comparable estimates of rates of child overweight/obesity [42]. In that study, the child’s BMI status did not influence the active parental consent procedure [42]. We cannot say with certainty why the participation rate was so much lower in the most recent survey. Besides the point discussed above regarding the increased public awareness of the topic, we have two potential explanations: (1) several teachers mentioned that more and more studies and surveys were conducted in schools in the past years, and parents have thus become less responsive, and (2) especially in schools with a high proportion of children with a migration background, parents may not be able to understand the participant information, which was only available in the national languages. The participant information sheets have become longer, giving more detailed information since the commencement of the new Human Research Act in 2014. This may have discouraged certain parents who have trouble reading and understanding the national languages.

In the current survey of 2017/18, we have not identified any statistically significant differences between the five regions Switzerland was divided into. This is in agreement with some, but not all, of the previous surveys. In 2002, we also did not identify significant differences in the prevalence between regions [26], while the southern region showed a significantly higher overweight prevalence in 2007 [24] and a higher obesity prevalence in 2012 (unpublished data). The southern region tended to have a higher overweight and/or obesity prevalence throughout the years, but as the surveys were not powered for the regional analysis and the participant numbers are small for this specific region, differences were not always significant.

As described earlier, obesity is a multifactorial problem, likely driven by a variety of apects [20]. In this study, we have only investigated those aspects that can be directly reported by the participants and their parents and not, e.g., early life risk factors, as it would have been even more difficult to receive reliable data on events 6 to 12 years in the past. In agreement with our previous study [21], we have identified foreign parental origin, low parental education, and low physical activity as predictors for overweight and/or obesity in children. Among children with both parents born outside of Switzerland, the odds for overweight and obesity was 1.7–2 compared to children with both parents born in Switzerland. Furthermore, in children with parents with low education, the odds for obesity was >3 compared to higher education. However, we have also shown that parents who were born outside of Switzerland tend to have a lower education compared to those born in Switzerland. Therefore, lower education might also be the main driver for an increased obesity prevalence here, rather than only cultural or genetic differences. Two regional studies carried out in Switzerland also identified low parental education, as well as foreign nationality or migrant status, as predictors for childhood overweight and obesity [43,44]. Studies in other countries, including Germany [45], the Netherlands [46], and Austria [47] also determined nationality or migrant status as important predictors for childhood obesity, but only the study in Austria also investigated and identified parental education as an additional predictor. In the most recent Health Survey for England (2017), it was shown that children living in the most deprived areas had a higher overweight and obesity prevalence compared to those from less deprived areas [48]. Another important predictor identified in this study was parental BMI, but the effect of migrant status was not reported [48]. Similarly, a Swedish study reported a higher prevalence of overweight and obesity in 10 year old children with a lower socioeconomic status based on neighborhood, but other influencing factors were not assessed [49]. Using a composite index of socioeconomic status, including education level, household wealth, and occupation status of the household head, a French study also reported higher odds for children being overweight or obese in the lowest compared to the highest group [50]. It was previously shown that education was the main driver for healthy dietary habits [51,52] in adults, which is likely to reflect children’s eating habits. On the other hand, children with mothers with a migrant status and with low education were also shown to eat more frequently in front of the TV, which was associated with a higher BMI, and higher intake of fat, carbohydrates, and soft drinks, but a lower intake of protein, fruits, and vegetables [53]. Similarly, a previous Swiss study also found an inverse relationship between TV viewing time and parental education [44]. At the same time, they found an association between TV viewing and the prevalence of overweight and obesity. All those studies highlight the importance of education and/or migrant status for lifestyle factors such as dietary habits and screen time. In our study, we have identified neither dietary habits nor screen time as a predictor for childhood overweight or obesity. However, those factors are much more sensitive to reporting errors compared to education and parental origin. Nevertheless, in our study, education and origin are also likely to influence both dietary habits and screen time and thereby overweight and obesity prevalence.

Weight gain and the development of obesity are primarily the result of a positive energy balance. In our study, none of the dietary factors assessed showed a significant prediction of overweight or obesity in the final model. However, we only asked about the frequency of consumption of certain food groups, and did not assess the overall energy intake. Furthermore, we were not able to include all dietary factors in the regression analysis due to small numbers for some of the fields, as described earlier. On the other hand, physical activity, measured as the number of days in the last week where the child was physically active for at least one hour, was a significant predictor, with an odds of 5 for activity on less than 1 day and an odds of 3.3 for 2–3 days compared to 6 and more days. In contrast, physical inactivity, in terms of screen time, was not a significant predictor in the final model combining boys and girls. In our previous study, screen time predicted overweight and especially obesity, predominantly in boys, and appeared more important than physical activity [21]. When looking at boys and girls separately in our current data set, screen time was predictive in boys, but the model could not be calculated in girls, due to small numbers. Our findings are comparable to an earlier HBSC (Health Behavior in School-age Children) survey (2001/2002), where physical activity and TV viewing, but not computer use, were important predictors of obesity [54]. Similarly, a Greek study found that overweight and obese adolescents were less physically active, but there was no difference in screen time [55]. While the association of low physical activity with overweight and obesity has a direct component through higher levels of energy expenditure, there are also more indirect benefits of physical activity during childhood, such as better motor development and higher confidence in one’s own ability [56]. As discussed above, increased screen time may not only reduce energy expenditure as it replaces more physical activities, but also tends to result in less healthy eating habits, especially if snacks or meals are consumed in front of the screen [44,53]. In our study, boys had a significantly higher prevalence of obesity and male sex was a significant predictor in the multiple regression model. This is in agreement with data from other European countries [57,58,59,60]. Therefore, it may be beneficial for prevention programs to specifically target boys.

Our study has several strengths, including the use of four nationally representative samples of school children, standardized measurement protocols throughout the years, and an extensive questionnaire to assess predictors in the two most recent surveys. Limitations of our study include the lower response rate in the 2017/18 survey, which may have increased non-responder bias. Additionally, the data used to identify risk factors was self-reported, and an accurate estimation of dietary factors and physical activity is challenging. Especially the dietary questions, which consisted of frequency questions over the past four weeks, may have been difficult to answer, especially for the younger children. This may be the reason why we did not identify any dietary risk factors in our study. Furthermore, we could not include all potential predicting variables in the models, as numbers for some of the answers were too small. Finally, socioeconomic status was only assessed using parental education, with parental origin as a further variable, but not including, e.g., household income, occupation, or living area.

In conclusion, future studies will be needed to confirm whether the small but significant decrease in childhood adiposity in Switzerland will continue. In the meantime, the prevalence of overweight and obesity at this age remains a public health concern, and our findings suggest prevention in schoolchildren might be most effective if focusing on migrant populations, as well as groups with a lower educational level. Furthermore, a special emphasis should be put on addressing boys, as their obesity prevalence was almost 50% higher than girls, and increasing physical activity.

## Figures and Tables

**Figure 1 nutrients-11-01922-f001:**
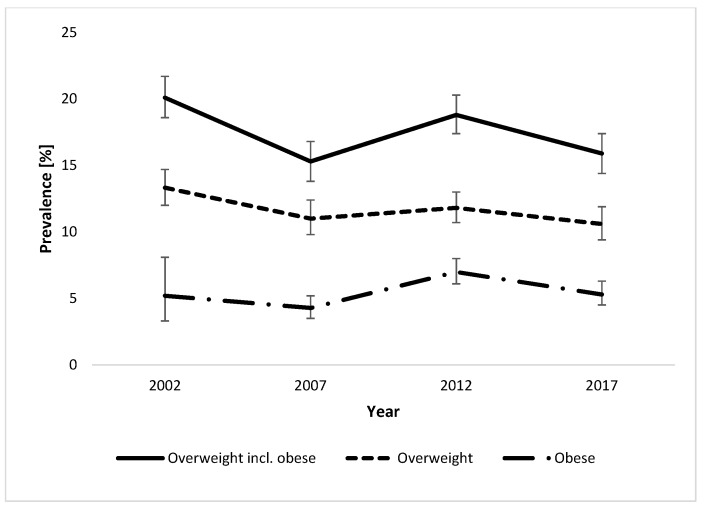
Trends in prevalence (95% CI) of overweight including obesity (all children with a body mass index (BMI) for age ≥85th percentile), overweight and obesity in Swiss children aged 6–12 years between 2002 and 2017/18, using the Centers for Disease Control and Prevention (CDC) reference. The prevalence of overweight incl. obesity and of overweight alone significantly decreased over time (OR= 0.988 (0.978–0.997) for overweight incl. obesity and (OR = 0.986 (0.975–0.997) for overweight alone), while the trend for obesity was not significant (OR = 0.994 (0.979–1.010)).

**Figure 2 nutrients-11-01922-f002:**
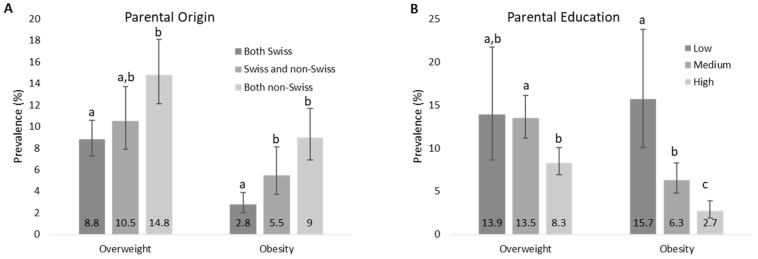
Prevalence (95% CI) of overweight (excluding obesity) and obesity in Swiss children aged 6–12 years in 2017/18 by parental characteristics (*n* = 2149). **A**: Prevalence by parental origin (country of birth), and **B**: prevalence by education level. Columns not sharing a letter within each block are significantly different from each other.

**Table 1 nutrients-11-01922-t001:** Characteristics of the study populations of national samples of Swiss children aged 6 to 12 years in 2002, 2007, 2012, and 2017/18.

	2002	2007	2012	2017/18
N	2493	2218	2963	2279
Sex (n (%))				
Boys	1231 (49.4)	1082 (48.8)	1499 (50.6)	1144 (50.2)
Girls	1262 (50.6)	1136 (51.2)	1464 (49.4)	1135 (49.8)
Age (y)	9.9 (6.2–13.0) ^a^	10.1 (6.3–13.0)	9.9 (6.3–13.0)	9.5 (6.0–12.9)
Weight (kg)	32.7 (17.7–94.4)	33.2 (15.9–83.3)	32.7 (16.7–132.3)	33.1 (16.7–106.2)
Height (m)	1.39 ± 0.120 ^b^	1.40 ± 0.116	1.39 ± 0.117	1.38 ± 0.111
BMI (kg/m^2^)	17.1 (12.5–35.0)	16.9 (12.3–34.7)	16.9 (12.4–42.7)	17.2 (11.9–42.5)
Body fat (%)	18.2 ± 9.0	-	19.3 ± 9.5	17.1 ± 8.0
Waist circumference (cm)	-	64.0 ± 8.0	63.2 ± 9.0	59.7 ± 7.1
Nr of schools	57	60	58	60
Response rate (%)	76	73	95	55

^a^ Median (min-max) (all such values); ^b^ Mean ± SD (all such values). BMI: body mass index; N: number. The response rate was calculated as the proportion of children who participated in the study out of all invited children in the participating schools.

**Table 2 nutrients-11-01922-t002:** Prevalence (% (95% CI)) of overweight and obesity (based on the Centers for Disease Control and Prevention (CDC) reference values) and elevated waist circumference in four national studies in 6 to 12 year old children in Switzerland between 2002 and 2017/18.

	2002 (*n* = 2493)	2007 (*n* = 2218)	2012 (*n* = 2963)	2017/18 (*n* = 2279)
**Overweight including obesity**				
Total	20.1 (18.6–21.7) ^a^	15.3 (13.8–16.8) ^b^	18.8 (17.4–20.3) ^a^	15.9 (14.4–17.4) ^b^
Boys	21.0 (18.8–23.4) ^a^	17.2 (15.1–19.6) ^b,c^	20.0 (18.1–22.1) ^a,c^	17.1 (15.1–19.4) ^b^
Girls	19.3 (17.2–21.5) ^a^	13.5 (11.6–15.6) ^b,^*	17.5 (15.6–19.5) ^a,c^	14.7 (12.8–16.9) ^b,c^
**Overweight**				
Total	13.3 (12.0–14.7) ^a^	11.0 (9.8–12.4) ^b^	11.8 (10.7–13.0) ^a,b^	10.6 (9.4–11.9) ^b^
Boys	13.4 (11.6–15.4) ^a^	11.8 (10.0–13.9) ^a^	12.1 (10.6–13.9) ^a^	10.8 (9.1–12.7) ^a^
Girls	13.3 (11.5–15.3) ^a^	10.2 (8.6–12.1) ^b^	11.5 (10.0–13.2) ^a,b^	10.4 (8.8–12.3) ^b^
**Obesity**				
Total	6.8 (5.9–7.9) ^a^	4.3 (3.5–5.2) ^b^	7.0 (6.1–8.0) ^a^	5.3 (4.5–6.3) ^b^
Boys	7.6 (6.2–9.3) ^a^	5.4 (4.2–6.9) ^b^	7.9 (6.7–9.4) ^a^	6.3 (5.0–7.9) ^a,b^
Girls	6. (4.8–7.5) ^a^	3.3 (2.4–4.5) ^b,^*	6.0 (4.9–7.4) ^a,^*	4.3 (3.3–5.7) ^a,b,^*
**Central adiposity**				
Total	-	12.0 (10.7–13.4)^a^	18.4 (17.0–19.8)^b^	6.3 (5.4–7.4) ^c^
Boys	-	11.5 (9.7–13.5)^a^	18.3 (16.5–20.4)^b^	7.3 (5.9–8.9)^c^
Girls	-	12.5 (10.7–14.6)^a^	18.4 (16.5–20.5)^b^	5.3 (4.1–6.8)^c^

* Significant difference between boys and girls for the respective year and category (z-test, *p* < 0.05). Values in each row not sharing a common superscript letter are significantly different between years (z-test, *p* < 0.05). Overweight and obesity were defined based on the 85th and 95th body mass index (BMI) for age percentile using the CDC reference values. Overweight including obesity includes all children with a BMI for age ≥85th percentile. Central adiposity was defined based on the 90th waist circumference percentile using Swiss reference values.

**Table 3 nutrients-11-01922-t003:** Risk factors for overweight and obesity in a national sample of school-aged children in Switzerland (*n* = 2149) analyzed using stepwise multiple regression.

	Healthy Weight	Overweight	Obese
%	%	OR (95% CI)	*p*	%	OR (95% CI)	*p*
**Parental origin**							
Both non-Swiss	23.3	35.8	1.695 (1.161–2.474)	0.006	47.1	2.037 (1.152–3.602)	0.014
Swiss and non-Swiss	19.8	19.5	1.236 (0.819–1.864)	0.313	22.1	1.754 (0.952–3.231)	0.072
Both Swiss	56.9	44.7	0		30.8	0	
**Parental education**							
Low	4.4	7.0	1.319 (0.662–2.630)	0.431	17.7	3.118 (1.458–6.666)	0.003
Medium	35.3	47.7	1.721 (1.245–2.377)	0.001	50.0	1.945 (1.181–3.204)	0.009
High	60.2	45.3	0		32.3	0	
**Physical activity**							
≤1 day	3.6	7.0	2.203 (1.058–4.588)	0.035	12.5	5.073 (2.083–12.355)	<0.001
2–3 days	23.2	36.7	2.238 (1.476–3.392)	<0.001	43.3	3.307 (1.755–6.231)	<0.001
4–5 days	39.5	34.5	1.387 (0.927–2.078)	0.112	28.8	1.393 (0.714–2.717)	0.331
≥6 days	33.6	21.8			15.4		
**Sex**							
Girls	50.5	49.0	0.824 (0.604–1.124)	0.222	40.5	0.554 (0.350–0.879)	0.012
Boys	49.5	51.0			59.5

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
