# Peer review of "Significant Decrease in Childhood Obesity and Waist Circumference over 15 Years in Switzerland: A Repeated Cross-Sectional Study"

_nutrients, 2019, doi:10.3390/nu11081922_

Round 1

Reviewer 1 Report

This paper uses a great data set and the overall findings (i.e. whether the prevalence of childhood obesity has reduced in Switzerland over time) is valuable.  However, in some areas, the paper would benefit from more development.  I have included comments/suggestions throughout the document attached, however key points to develop/improve reader understandability are:

·         Presentation of the results in the tables/figures could be more reader friendly – it was confusing to look for superscript letters between columns/figures etc

·         The discussion could be more in depth and provide more insight as to the “why” behind the trends observed. The co-authors broadly describe other studies who have found similar findings, however do not provide any insight as to the why behind these factors influencing ow/ob.  For example, why might parental migrant status influence the child’s risk of ow/ob? Genetics? Environment? Culture? Norms? This paper needs to address the bigger picture implications of these findings – what does it mean?  How do we tailor our approaches to improve ow/ob based upon your findings?  Have others done this, and how effective was it?  How do we improve parental education?  How do we increase physical activity in children? How do we specifically target boys? The data presented is current and great leverage points to begin with, but it’s very descriptive and lacking exploration and insight into the findings and bigger picture implications of the work.

·         Not enough emphasis is made on the fact that the current prevalence of ow/ob, regardless of whether it has slightly reduced over time, is still too high!  Does Switzerland currently have a national obesity strategy or initiative?

·         Avoid using “normal weight” as this could be potentially offensive, use “healthy weight”

Reviewer 2 Report

The authors present descriptive data on the prevalence of overweight and obesity in school aged children in Switzerland using data from national surveys in years 2002, 2007, 2012 and 2017/2018. Moreover authors try to identify potential determinants of obesity among children participating in the 2017/2018 survey. The main strength of this paper is the study design and the availability of anthropometric data from different generations of children for over 15 years in Switzerland. Despite, the manuscript (including abstract) lacks to provide a clear, adequate and comprehensive presentation of the research hypothesis and results. My concerns/comments and questions are addressed as follows:

Broad comments

1)    Text structure does not facilitate the reader to understand the scientific background, the rationale for conducting this study and does not provide a focused interpretation of the findings of this study. For example in the introduction authors spend a whole paragraph on NCDs. I would prefer to read more details on the trends in childhood obesity worldwide, in Europe and in Switzerland as well as possible hypotheses for these changing trends. Moreover, the paragraph on the predictors of obesity is very narrow and not at all focused to the research interest of this Swiss study. I would recommend that introduction and discussion should be rewritten cautiously.

2)    Introduction. Last paragraph. “The secondary objective was to assess predictors for the development of NCDs using a self administered questionnaire.”. Which NCDs? I understood that authors look only on overweight/obesity.

3)    Methods line 91. Could authors denote whether collection of data in previous years was different/similar to that described for the years 2017/2018?

4)    Methods line 158. Not the correct reference (this is just another paper using the same equation for BF%). 

5)    Methods line 165, are there any Swiss reference curves for BMI?

6)    Methods line 170, it is not clear to me why authors combined overweight and obesity (overweight including obesity) but they present also results only for obesity. Maybe it is better if you will use a different term such as “overweight or obese” because “overweight including obesity” is misleading.  Try to be consistent regarding terms throughout the text.

7)    Results. Why do authors present only info on the 2017/2018 survey? I would expect to see some info on the other surveys as well even if this is published before. Could you sum up all this info in a flow diagram?

8)    Lines 217, 264. This is misleading. Authors should simply report “WC >90thpercentile” instead of “metabolic disease”.

9)    Line 266. Could authors report regional differences in overweight/obesity in surveys in 2002, 2007 and 2012?

10)Table 1, please include statistics for the four skinfolds.

11)Line 342. Why do you believe you had such a low participation rate in this survey compared to the others?

12) Abstract line 22 and discussion Line 397. Please avoid such definite statements. This study lacks info on important early life predictors of obesity such as breastfeeding and so this study alone cannot provide public health recommendations.

13)Reporting of this study is not detailed and clear enough. I would suggest that the authors follow the STROBErecommendations in order to improve their paper and facilitate the interpretation and application of study results. This checklist developed by Vandenbroucke et al along with an explanatory and elaboration document are published at several journals with free access.

14)Were there children with specific health problems that were excluded? Were there any outliers? Were there children with missing data on one of the anthropometric measures? How did authors handle with missing data? Were there obese children with too large skinfold thicknesses that was impossible to measure with caliper? Regarding measurement error it is known that TEM is age dependent and related to the anthropometric characteristics of the group or population under investigation. Try to calculate this (line 109). Check for more details the publication by Stanley J. Ulijaszek and Deborah A. Kerr (Anthropometric measurement error and the assessment of nutritional status). 

15)Table 3. Normal weight includes underweight children as well or those are excluded?

Specific comments

1.     Make sure that you provide all abbreviations. E.g. Introduction, line 39, OECD abbreviation 

2.     Methods line 130, please replace media consumption with screen time.

3.     Line 206, is this a headline? “Trends in overweight and obesity prevalence between 2002 and 2017/18 

4.     Results lines 236-258. If it is ok with journal requirements please include in parentheses only OR with CI. 

Round 2

Reviewer 1 Report

I have reviewed the updated manuscript and response to reviewer document.

 I still believe the tables (e.g. Table 2) are a little confusing i.e. the “values in each row not sharing a common superscript letter….” specifically the a,A,B…..what does a stand for, and A? And B?  What are the differences?  Can this not be presented in a different way?

However I understand that there is a lot of data to present here.

Author Response

 I still believe the tables (e.g. Table 2) are a little confusing i.e. the “values in each row not sharing a common superscript letter….” specifically the a,A,B…..what does a stand for, and A? And B?  What are the differences?  Can this not be presented in a different way?

However I understand that there is a lot of data to present here.

Our reply: Thank you for this comment. We agree the table was not yet ideal and we have tried to make it easier to understand. We have removed the columns at the right for the comparisons between sex again and have introduced * directly behind the values of the girls if there was a significant difference. We also thought about the superscript letters. However, we could not come up with a simpler solution. If we have to use a different symbol for each comparison (like, 2002 vs. 2007, 2002 vs. 2012, etc.) and explain those in the footnotes the table would only get more confusing. So we would like to present the comparisons using the superscirpt letters. In a row, if two values do not share a superscript letter, they are significantly different. So for Overweight, year 2002 has only an a, while 2007 has only a b. So they are different. 2012 is not different from either 2002 or 2007 as it shows a,b.  Also, we have realized, that there still was a mistake in the table, as we forgot to delete two lower case letters that remained from the comparisons between sex. Also, we have moved the superscript letters behind the confidence interval which seems to improve readability and have changed from capital letter to lower case letter. We hope those changes make the table easier to understand.